# OpenReview forum: "AMGE: Adaptive Modality Gap Exploitation for Adversarial Attacks on Vision-Language Models"
_TMLR — Rejected by TMLR_

### Review · Reviewer_UKUk · 2025-11-08

**Summary Of Contributions:**

The paper proposes AMGE (Adaptive Modality Gap Exploitation), an adversarial attack on multimodal LLMs that explicitly exploits the modality gap between image and text embeddings. It estimates a gap direction between modalities, aligns gradients with this direction (including via cross-attention), and learns universal perturbations (sometimes over model ensembles) that preserve image semantics while inducing harmful responses. Experiments on open MLLMs (e.g., LLaVA, Qwen-VL, Qwen2-VL) show high attack success and some cross-model generalization; the paper also includes a stylized theoretical analysis linking gap geometry to attack success.

**Audience:**

Yes

**Audience Explanation:**

- The paper presents a clear new angle: Treats the modality gap as an attack surface, not just operating blindly in pixel space.

- Methodologically rich: Gap direction estimation, gap-aligned loss, cross-attention–mediated gradients, universal perturbations with ensemble training and centroid regularization.

**Broader Impact Concerns:**

The paper substantially advances the effectiveness and transferability of multimodal jailbreak attacks in a gray-box setting, which raises clear dual-use concerns. By explicitly exploiting modality gaps and providing detailed attack algorithms and settings, the paper could lower the barrier for malicious actors to bypass safety filters of deployed vision–language systems, potentially enabling harmful applications (e.g., violence, harassment, illegal activities) at scale. The ethical implications would need a Broader Impact Statement that (1) explicitly acknowledges this dual use, (2) justifies the level of technical detail released, and (3) discusses concrete mitigation strategies (e.g., how insights from modality-gap exploitation can inform robust defenses, responsible release practices, and coordination with model providers).

**Claims And Evidence:**

No

**Claims Explanation:**

-  Unfair comparisons. AMGE assumes encoder representation + gradient access (gray-box), which is much stronger than typical real-world black-box API access. The comparison to MML-Attack is not threat-model aligned: MML is originally a black-box, prompt-/structure-based attack on GPT-4-class APIs, whereas AMGE’s setting allows gradients on open models. Their lower “MML-Attack” numbers likely reflect a modified reimplementation under different models/datasets/evaluation, so cross-paper claims like “better than MML” are not fully trustworthy.

- Transferability framing. What the paper calls “transferability” is mainly ensemble-trained universal perturbations that generalize across models included (or related to those) in the ensemble. This is different from classical transfer where you attack a single source model and test unchanged perturbations on unseen targets. The paper should emphasize this as ensemble-based generalization, not standard transfer.

- Theoretical issues. Theorem 1 invokes a Lipschitz constant for the LLM / decision boundary, but the final bound does not depend on the Lipschitz constant, and the proof never actually uses it; the assumption is effectively redundant or inconsistent with the stated result. The analysis relies on strong simplifications (1-D Gaussian noise along the gap, linear margin behavior). It’s useful intuition but not a robust guarantee for real high-dimensional MLLMs; this should be presented more modestly.

- Experimental clarity & ablations. Many moving parts (gap-aligned term, cross-attention gradient, universal centroid regularizer, ensemble training), but no thorough ablation to show which are critical vs. minor. Evaluation is exclusively on open models with encoder access, with no demonstration on closed APIs even in a purely query-based (no-gradient) setting.

**Requested Changes:**

Please see above weakness points.

---

### Review · Reviewer_5SDY · 2025-11-23

**Summary Of Contributions:**

The paper proposes Adaptive Modality Gap Exploitation (AMGE), a gray-box adversarial attack framework that exploits the geometric "modality gap" between vision and text embeddings as a vulnerability surface. By utilizing cross-attention-mediated gradient flow and momentum-based ensemble aggregation , the method achieves high attack success rates and query efficiency on LLaVA and Qwen-VL series models.

**Audience:**

Yes

**Audience Explanation:**

The paper addresses the timely and critical topic of adversarial robustness in Multimodal Large Language Models (MLLMs). The specific proposal to exploit the "modality gap" as a safety vulnerability connects representation learning dynamics with AI safety in a novel way. This geometric perspective on jailbreaking would be of significant interest to researchers working on multimodal alignment, embedding space analysis, and adversarial defense, regardless of the experimental limitations noted regarding the specific execution.

**Claims And Evidence:**

No

**Claims Explanation:**

The claims of stealthiness and comparative superiority are unconvincing, as the experiments utilize a perturbation budget ($\epsilon=16/255$)  significantly exceeding the standard threshold for imperceptibility. In addition, the paper fails to specify the budgets used for baselines; if they were evaluated at the standard $\epsilon=8/255$ while AMGE used $16/255$, the comparison is fundamentally unfair. Secondly, the theoretical lower bounds provided are also loose, relying on circular conditions satisfied by the high budget rather than fundamental geometric exploitation. Finally, the claim of universal transferability  remains unproven due to the lack of evaluation on architectures with information bottlenecks, such as InstructBLIP's Q-Former, which could filter out the proposed perturbations.

**Requested Changes:**

- Paper format issues:
    - Rescale Table 2 and Table 3  to fit within the page margins.
    - Add explanatory text to Section 5.1 (currently empty) to justify the model selection.
    - Fix Figure 4  where the "Transferability Score" label obstructs the heatmap data.

- Validate Stealthiness at Standard Budgets: To substantiate unsupported claims of stealthiness given the non-standard perturbation budget of $\epsilon=16/255$, the authors should either re-evaluate the method at the standard $\epsilon=8/255$ or provide a rigorous study verifying that the adversarial artifacts remain imperceptible.
- Clarify Baseline Settings for Fairness: The submission does not specify the perturbation budgets used for the baseline methods (MML-Attack, PolyJailbreak, etc.) in Table 4.
- Evaluate on Architectures with Information Bottlenecks: To substantiate the claim of "universal transferability," the evaluation must extend beyond LLaVA/Qwen (which use direct projection/MLP) to include architectures with stronger information bottlenecks, specifically InstructBLIP (which uses a Q-Former).
- Revise Theoretical Claims: The discussion of Theorem 1 must be revised to explicitly acknowledge that the bound relies on the circular condition $m \cdot \epsilon \ge 2 \Delta$, which essentially guarantees success through brute-force perturbation magnitude ($\epsilon=16/255$) rather than specific geometric properties, rendering the bound loose.

---

### Review · Reviewer_iDc8 · 2025-12-01

**Summary Of Contributions:**

Key contributions:
1. A method to empirically estimate a directional gap vector ('Modality Gap' vector) from paired data, which serves as a guide for adversarial perturbations.
2. A gray-box attack that uses a surrogate model to define a specific optimization goal: finding the smallest image perturbation that pushes the visual embedding along the 'Modality Gap' vector. The algorithm solves for a noise pattern that forces the image to cross the geometric boundary separating vision from text, effectively making it land in the 'harmful text' region of the feature space.
3. A momentum-based ensemble strategy that generates perturbations capable of transferring effectively to black-box target models even if they have different architectures.
4. The authors provide theoretical bounds linking the modality gap magnitude to attack success probability.
5. The approach has strong results: a 90.2% Average Attack Success Rate and 79.1% Transferability across diverse architectures, while requiring significantly fewer queries (~127) than competing methods.

Strengths:
1. Shifting the attack surface from pixel/token optimization to the embedding space is very interesting and seemingly effective innovation proposes in this paper.
2. The method is much more query-efficient than black-box baselines, making it a more practical.
3. The empirical results show that the modality gap is a consistent, transferable vulnerability across different model families.

Weaknesses:
1. The evaluation compares AMGE (a gradient-based surrogate method) primarily against query-based/black-box attacks. A comparison against a standard Transfer-based PGD baseline would isolate the specific contribution of the gap-vector constraints which is a core feature of the approach in this paper. It is unclear if to me if this is covered in the provided baselines.
2. The paper lacks evaluation against standard input transformations (JPEG compression, resizing, etc.), which are common in real-world API deployments and could potentially neutralize this attack.

**Audience:**

Yes

**Audience Explanation:**

This work is likely to interest researchers working on architectures or adversarial attacks in vision, language, and just LLMs in general as they become more and more multimodal. Additionally, the findings regarding the transferability of these perturbations provide interseting insights into the shared vulnerabilities of LLMs -- which is highly relevant for those studying LLM safety.

**Broader Impact Concerns:**

This paper may need some kind of Broader Impact Statement since it presents an effective jailbreak capable of bypassing safety guardrails in multimodal LLMs to generate harmful content. This method could be utilized by malicious actors to automate the bypass of safety filters in LLMs.

**Claims And Evidence:**

No

**Claims Explanation:**

While the experimental results show high attack success rates (ASR), the evidence does not yet convincingly support the claim that this is mainly due to the the proposed algorithm. The submission compares AMGE (which utilizes gradients from a surrogate model) primarily against query-based or black-box attacks. To convincingly prove that the proposed geometric constraints are better than standard approaches, the authors should compare AMGE against a projected gradient descent (PGD) baseline optimized on the surrogate without the gap-vector or attention constraints. If VLAttack satisfies this, more explanation and details would greatly help.

Additionally, the claim that this represents a practical "fundamental vulnerability" is not fully supported without evidence of robustness to standard input transformations. The submission lacks data on whether the embedding-space perturbations survive basic defenses like JPEG compression or resizing.

**Requested Changes:**

1. To validate the core scientific claim, that exploiting the modality gap yields superior performance, you should compare AMGE against a standard gradient-based baseline, not just query-based or black-box attacks. Please add a comparison against a standard PGD attack optimized on the surrogate model using the same budget. If VLAttack satisfies this, can you explain?

2. Since the paper frames AMGE as a practical jailbreak threat, you should demonstrate that the attack survives basic, non-adversarial defenses common in real-world API deployments. Please provide experimental results showing the Attack Success Rate (ASR) when the adversarial images are subjected to JPEG compression and/or Resizing/Downscaling before reaching the target.

3. To better isolate the sources of improvement, it could be interesting to include an ablation study where the cross-attention term is removed. Additionally, I think a brief discussion hypothesizing why attention gradients from a surrogate transfer effectively to different architectures would be valuable.

---

### Comment · Action_Editor_c3gC · 2025-12-14
**Rebuttal**

Dear Authors,

This is a kind reminder that the rebuttal period has started for a while and we haven’t received your response. Are you planning to submit a rebuttal? The reviewers will be asked to submit their final recommendations soon. Thanks!

Best, AE

---

### Decision · Action_Editor_c3gC · 2026-01-07

**Recommendation:** Reject

**Audience:**

Yes

**Audience Explanation:**

The topic of adversarial robustness in Multimodal LLMs is timely and significant. The specific angle of analyzing the "modality gap" as a security vulnerability offers a novel geometric perspective that would interest researchers in multimodal alignment, embedding space analysis, and AI safety.

**Claims And Evidence:**

No

**Claims Explanation:**

While the reviewers agreed that the core idea of exploiting the modality gap is interesting, there is a consensus that the claims are not supported by the evidence provided.

- Unfair Baselines: The method (gray-box, utilizing gradients/encoders) is compared primarily against black-box or query-based attacks. Reviewers noted the lack of a proper gradient-based baseline (e.g., PGD on a surrogate model) to isolate the benefits of the proposed gap exploitation.

- Perturbation Budget: The experiments utilize a perturbation budget ($\epsilon=16/255$) that significantly exceeds standard thresholds ($\epsilon=8/255$), undermining claims regarding stealthiness and fairness of comparison against baselines.

- Theoretical Concerns: The theoretical bounds presented were described as loose or relying on circular conditions that are satisfied by the high perturbation budget rather than the geometric properties of the method.

- Lack of Robustness Checks: There is a lack of evaluation against standard input transformations (e.g., compression, resizing) and architectures with information bottlenecks (e.g., InstructBLIP).

Crucially, the authors did not submit a rebuttal to address these significant methodological flaws raised during the review process.